# Serological Responses in Cattle following Booster Vaccination against Serotypes 4 and 8 Bluetongue Virus with Two Bivalent Commercial Inactivated Vaccines

**DOI:** 10.3390/v14122719

**Published:** 2022-12-05

**Authors:** Corinne Sailleau, Lydie Postic, Xavier Chatenet, Olivier Salat, Mathilde Turpaud, Benoit Durand, Damien Vitour, Stéphan Zientara, Emmanuel Bréard

**Affiliations:** 1UMR VIROLOGIE, INRAE, École Nationale Vétérinaire d’Alfort, ANSES Laboratoire de Santé Animale, Université Paris-Est, 94700 Maisons-Alfort, France; 2Inovet, Orion, 3 rue de Lorraine, 62510 Arques, France; 3Commissions Vaches Laitières et Qualité du Lait de la SNGTV, Clinique Vétérinaire de la Haute Auvergne, 15100 Saint Flour, France; 4Epidemiology Unit, Laboratory for Animal Health, French Agency for Food, Environmental and Occupational Health and Safety (ANSES), University Paris-Est, 94700 Maisons-Alfort, France

**Keywords:** bluetongue, vaccination, serological analysis

## Abstract

Since the outbreak of bluetongue in Northern Europe in 2006, numerous outbreaks involving several serotypes have been observed. Since 2008, compulsory or voluntary vaccination campaigns with inactivated vaccines have been carried out to eradicate these serotypes. In France, serotypes 8 and 4 have been enzootic since 2017, and currently, the majority of vaccinations take place in the context of animal movements, to comply with the regulations of the importing countries. Several vaccine manufacturers have developed inactivated vaccines against serotypes 4 and 8 (mono or bivalent). In this study, we investigated and compared the serological responses to a booster vaccination with two different bivalent inactivated vaccines (BTVPUR suspension injectable^®^ 4 + 8, Boehringer Ingelheim or SYVAZUL ^®^ BTV 4 + 8, Biové) following a primary vaccination with BTVPUR^®^ 4 + 8 in the previous year. The results show that using an alternative vaccine for booster vaccination is at least as effective as using the homologous vaccine. Indeed, the antibody response against BTV-8 is higher in the case of a heterologous vaccination and identical for BTV-4. This information could allow more flexibility in the choice of vaccines used for booster vaccination, particularly in cases where homologous vaccines are in short supply or unavailable.

## 1. Introduction

Bluetongue (BT) is a vector-borne disease affecting domestic and wild ruminants, caused by the BT virus (BTV) that can induce significant losses in ruminant production, mainly in sheep, as cattle and goats generally remain asymptomatic [1]. Indeed, the BTV-8 strain present in Europe since 2006 is one of the strains capable of inducing clinical signs in cattle due to its pathogenic and transplacental transmission properties [2].

In Europe, before the 2000s, BTV outbreaks were sporadically observed in the Mediterranean Basin. Subsequently, the global distribution of BTV has changed, with continuous emergence particularly in Europe and in the Mediterranean Basin. These BTV incursions have involved different serotypes: BTV-1, 2, 3, 4, 6, 8, 9, 11, and 16. In France, five serotypes have circulated over the last 20 years, with sometimes several serotypes present at the same time and in the same place, as observed in France in 2008–2010 (BTV-1 and 8) and since 2018, when France shifted from a BTV-free status in 2000 to an enzootic status for serotypes 4 and 8 [3].

In response to the different BTV incursions in France, the French Veterinary Authorities set up several vaccination campaigns. Live-attenuated vaccines were first used (in Corsica between 2000 and 2004) because they were initially the only type of vaccine widely available commercially. The use of this type of vaccine has revealed several drawbacks such as the induction of clinical signs due to the under-attenuation of the virus, reduced milk production, abortions in pregnant females, reassortment with field strains, etc. Subsequently, BTV vaccination performed in Europe, during compulsory or voluntary campaigns, was only carried out with inactivated vaccines [4] produced by several manufacturers.

The commercially available BTV inactivated vaccines have shown good safety and efficacy in both cattle and sheep, and studies have shown that they are protective for at least one year post-vaccination [5,6,7,8,9]. Thus, after the primary vaccination with an inactivated vaccine, it is recommended to conduct a booster the following year. In countries where the use of multiple inactivated vaccines against a BTV serotype is authorized, farmers may have an interest in being able to perform annual boosters with different vaccines than the one used for the primary vaccination. In some circumstances, vaccines may be unavailable or no longer available, or their cost may change.

To our knowledge, only one study was performed in 2011 showing that ruminants primed with a monovalent BTV-8 inactivated vaccine and boosted the following year with an alternative vaccine had neutralizing antibody levels at least as high as with the homologous vaccine [10]. The aim of our study was to verify that the use of an alternative BTV commercially inactivated vaccine (as a booster) allows the induction and/or increase of protective antibodies similar to a boost with the homologous vaccine. This study was carried out with bivalent BTV vaccines against BTV-4 and -8 available in 2022. These two vaccines BTVPUR^®^ 4 + 8 (Boehringer Ingelheim) and SYVAZUL^®^ BTV 4 + 8 (Syva) have received European marketing authorization (in 2016 and 2019, respectively) and are widely used in Europe and the Middle East.

## 2. Materials and Methods

### 2.1. Animals

The whole study was conducted with 50 cattle, from 3 farms (A, B and C), all primo-vaccinated in 2021 against BTV with a bivalent inactivated vaccine against serotypes 4 and 8 (BTVPUR^®^ 4 + 8—Boehringer Ingelheim, Ingelheim am Rhein, Germany). In farms A and B (located in Cantal), 19 Montbeliarde cows and 18 Holstein cows were included in the trial, respectively. Thirteen Montbeliarde cows from farm C, located in Haute-Loire, were also included.

### 2.2. Study Design

The date of the vaccine booster in 2022 is day 0 (D0) of the experiment (Table 1). On this day, EDTA and whole blood (serum) samples were collected from the cattle included in the study before administering the booster vaccination with BTVPUR^®^ 4 + 8 “BTVPUR” or SYVAZUL^®^ BTV 4 + 8 “SYVAZUL”, performed by the herd’s veterinarian, following the suppliers’ recommendations. All animals did not demonstrate clinical signs. EDTA samples were then taken 10 days post-vaccination (D10) and finally, at D42 (farm C) or D56 (farm A and B). Whole blood (serum) samples were collected at D42 in all farms. For farm C, only 10 of the 13 animals were sampled at D42 (4 vaccinated with BTVPUR and 6 with SYVAZUL). Thus, the total number of animals tested at D0 was 50 and at D42 was 47. Indeed, 24 animals received the SYVAZUL vaccine and 23 received the BTVPUR vaccine for the annual booster.

### 2.3. Real-Time RT-PCR (rtRT-PCR)

Total RNA was extracted from 100 µL of EDTA blood sampled at D0, D10, and D42 or D56 using the Kingfisher 96 robot and the ID Gene Mag Universal Isolation kit (Innovative Diagnostics) according to the manufacturer’s instructions. Finally, RNAs were eluted in 80 µL of ultrapure water and 5 µL was used in a specific BTV reverse-transcription polymerase chain reaction (RT-PCR). rtRT-PCR was performed using a commercial kit (ADI–352, BioX) according to the manufacturer’s instructions.

### 2.4. ELISA

The whole blood samples were centrifuged at 2500 g for 5–10 min to obtain the serum. The anti-VP7 BTV antibody levels were estimated from sera sampled at D0 and D42 using the cELISA ‘ID Screen^®^ Bluetongue Competition’ assay (Innovative Diagnostics, Montpellier, France) according to the manufacturer’s instructions. Sera with an inhibition percentage < 35 % were considered as positive.

Sera were also tested by an in-house double antigen sandwich ELISA able to specifically detect VP2 BTV4 [11]. The ELISA protocol used was the same as described in the article, with one modification: the ELISA buffer used was no longer the Innovative Diagnostic buffer #14, but PBS-Tween (0.5%) − Milk (5%) − BSA (1%). The sera were considered positive when the Optical Density (OD) value was > 0.25.

### 2.5. Serum Neutralization Test (SNT)

Sera sampled at D0 and D42 were titrated for the presence of serotype-specific BTV-4 and -8 neutralizing antibodies by SNT. Briefly, two-fold dilutions of sera (in duplicate) were performed in a Minimum Essential Medium (MEM, GIBCO^®^, Waltham, MA, USA), starting at 1/5 (range of dilution: 5–640). Fifty microliters of MEM, containing 100 TCID50 of each BTV serotype, were incubated in microtiter plates, with 50 µL of the diluted sera for 1 h at 37 °C. Then, 20,000 Vero cells in 100 µL of MEM supplemented with 10% fetal calf serum and 2% sodium pyruvate were added in each well. The microtiter plates were then incubated for 6–7 days at 37 °C. The neutralizing titer of each serum was defined as log(2) of the inverse of dilution allowing 50% neutralization of the 100 TCID50 (range of reading 2.32–9.91).

### 2.6. Statistical Analysis

We used the Student’s T-tests to compare the mean titer of neutralizing antibodies at D0 and D42, for the two vaccines used as boosters, and for the two viruses. We used two linear mixed-effects models (for BTV4 and BTV8, respectively) to analyze the effect on the D42 SN titer of the vaccine used for booster vaccination. In both models, the dependent variable was the D42 SN titer, and the fixed effects were the SN titer at D0 (quantitative variable) and the vaccine used for booster vaccination (qualitative variable: BTVPUR or SYVAZUL, BTVPUR being the reference value). The farm was treated as a random effect. The statistical analysis was performed using R and the lme4 package [12,13].

## 3. Results

All animals were BTV RT-PCR negative at D0, 10, and 42 (farm C) or D56 (farms A and B) (data not shown).

Prior to the booster vaccination (D0), 78% of the cattle were positive according to the VP7 ELISA, while 82 and 98% were detected positive according to the BTV4 and BTV8 SNT, respectively (Table 2) (range of results: 2.32–7.91 and 2.32–8.32 for BTV-4 and BTV-8, respectively). At D42 after the booster, all the 47 cattle tested were both positive according to the VP7 ELISA, with a % S/P ≤ 4 (data not shown) and the SNT BTV-4 and BTV-8 (range of results: 5.32–9.32 and 6.32–9.91 for BTV-4 and BTV-8 respectively).

The sera of 46 animals collected at D0 and D42 were tested by VP2 BTV-4 ELISA. The results of the BTV4 ELISA showed that at D0, seven sera were negative (Figure 1). After vaccination with BTVPUR or SYVAZUL, surprisingly, 1 serum out of the 46 tested remained negative at D42. On average, the OD increased between D0 and D42 (at D0: 1.05 +/− 0.60 to 1.66 +/− 0.64 with BTVPUR, and 1.67 +/− 0.44 with SYVAZUL).

Regarding the neutralizing antibodies (Nab) titers, an increase was observed in each farm between D0 and D42 whatever the vaccine used as a booster and for the two viruses (Table 3 and Figure 2). The statistical analysis confirms that this increase is significant (Student’s T-test for paired samples: *p* < 0.0001 in both cases).

Concerning BTV4, the mixed-effects model showed that using the SYVAZUL vaccine for booster vaccination instead of BTVPUR did not induce any significant effect on the D42 SN titer (*p* = 0.07). The D0 SN titer was significantly and positively associated with the D42 titer (*p* = 0.02).

For BTV8, using the SYVAZUL vaccine for booster vaccination instead of BTVPUR had a significant effect on the D42 SN titer (*p* < 0.0001), which was increased by 1.03. No effect of the D0 SN titer was observed.

## 4. Discussion

The results of the PCR assays performed on samples collected at D0 and D42 (farm C) or D56 (farm A and B) allowed the exclusion of BTV circulation in the farms during this study.

During past surveillance programs, the question of a possible interference of the vaccination with the detection of low loads of BTV RNA in some animals by real-time RT-PCR assays was raised [14,15]. Blood samples were collected at D10 post vaccination to evaluate the possibility of detecting BTV RNA derived from vaccines with the rtRT-PCR used in this study.

The most recent studies performed on vaccinated sheep or cattle have given contradictory results on the detection or non-detection of RNA following vaccination with an inactivated vaccine [9,16,17,18]. Our rtRT-PCR assays on 50 blood samples collected at D10 did not detect any BTV RNA. The detection or non-detection of RNA from the vaccines described in these investigations can be explained by the heterogeneity of the protocols used. Previously published studies have all used monovalent BTV-8 vaccines (BTVPUR^®^ AlSap 8 or Bovilis-BTV-8). The real-time RT-PCRs used targeted different genes (segment 1, 5, or 10) [19,20,21,22] and were performed on samples taken at different times after vaccination. In the studies where vaccine RNA was detected [16,17,18], it was of very low loads (Threshold Cycle > 35).

The VP7 ELISA results show that this test can detect 78% of primed animals positive one year after their primary vaccination with a bivalent inactivated vaccine. Zanella et al. [23] have already observed similar results in a vaccination follow-up in 18 cattle: eight were VP7 ELISA negative 335 days after their primary vaccination with BTV inactivated vaccines. These results are fully in line with the vaccine manufacturers’ recommendations regarding the need for an annual booster. The detection of anti-VP2 antibodies seemed to be the best strategy to demonstrate the presence of antibodies induced by inactivated vaccines. In our study, the most sensitive tests seemed to be the SNT and the VP2 BTV-4 ELISA. At D0, 82 and 98% of the animals were detected positive according to the SNT BTV-4 and -8, respectively (Table 2). Concerning the VP2- BTV4 ELISA, 84.7% of the 46 animals tested were positive at D0. After their booster, 45 animals were VP2 BTV-4 ELISA positive at D42. It is not clear why one of the cattle, which received a BTVPUR vaccine, remained VP2 BTV-4 ELISA negative when it was positive according to the SNT BTV-4 and VP7 ELISA.The sensitivity of this VP2 BTV-4 ELISA is lower than the SNT (11). Thus, this negative result could be due to this lack of sensitivity and would be a false negative result. Taken together, these data show that type-specific ELISAs can be used to assess a serological response induced by inactivated vaccines.

The difference in antibody detection results between tests is not incompatible because the target proteins are not the same. Inactivated vaccines, consisting of virus particles, have on their surface VP2 against which serotype-specific and neutralizing antibodies are directed. The VP7 protein is located under the outer capsid (VP2 and VP5) and is less accessible to the immune response. Thus, it can be assumed that if vaccine-derived antibodies to VP7 are still present one year after vaccination, they may be present in lower amounts than antibodies to VP2. After the annual boost, the VP7 ELISA detected all animals with a % S/P ≤ 4, equivalent to a strong positive result. It may be interesting to test these animals in one year to observe or not the maintenance of these antibodies anti-VP7 levels.

Regardless of the vaccine used for the booster vaccination, all animals were positive at D42 with tests commonly used to assess the immune response after vaccination (ELISA VP7, SNT BTV-4 and 8). Moreover, an increase in neutralizing antibody titers against BTV-4 and BTV-8 was observed (Table 3, Figure 2). The difference between the overall mean titers at D0 and D42 (2.66 (serotype 4), 2.16 (serotype 8), 3.08 (serotype 4), and 3.50 (serotype 8)) for the BTVPUR and SYVAZUL vaccines, respectively, (Table 3) corresponded to an increase in antibody titers by a factor of 6 (serotype 4) and 4 (serotype 8), and 8 (serotype 4) and 11 (serotype 8) for the BTVPUR and SYVAZUL vaccines, respectively. Indeed, as observed by Bartram et al., 2011, a booster performed with another vaccine than the one used for the primary vaccination is not only as effective as if the booster vaccine is the same type as the primary vaccination, but even seems to induce a higher increase of Nab against the serotype 8 (the vaccines used in their study were two BTV-8 monovalent vaccines). In our study, the statistical analysis showed that the increase in BTV-8 titer induced by the SYVAZUL vaccine was significantly higher (*p* < 0.0001) when compared with the increase in BTV-8 titer induced by the BTVPUR vaccine (*p* = 0.07).

In our study, two bivalent vaccines, BTVPUR^®^ 4 + 8 and SYVAZUL ^®^ BTV 4 + 8, were used as booster doses after a primary vaccination one year earlier with BTVPUR^®^ 4 + 8. An additional assay would be interesting to assess whether the inversed protocol (a primary vaccination with SYVAZUL followed by a booster with BTVPUR) would give similar results. The results obtained by Bartram et al., as well as ours, suggest that, in cattle, a booster with an alternative vaccine induces an increase in neutralizing antibodies at least as effectively as after a booster with a homologous vaccine.

In conclusion, the cross-use of vaccines from different manufacturers for primary and booster vaccination is not only possible, but could be recommended. Further to this, the option to use heterologous vaccines as a booster means that this provides veterinarians more freedom and flexibility in situations where vaccines are in short supply or unavailable.

## Figures and Tables

**Figure 1 viruses-14-02719-f001:**
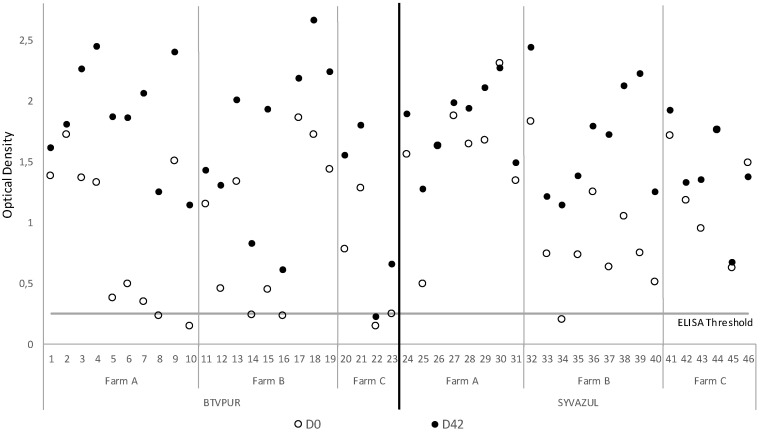
Double-antigen sandwich ELISA serotype 4-specific antibodies OD results from sera collected prior to D0 (white circle) and post D42 (black circle) booster vaccination with either BTVPUR or SYVAZUL. X axis represents the animals boosted with BTVPUR (1–23) and SYVAZUL (24–46). Horizontal line represents the cut-off value (threshold). Circles crossing the line are considered as negative results.

**Figure 2 viruses-14-02719-f002:**
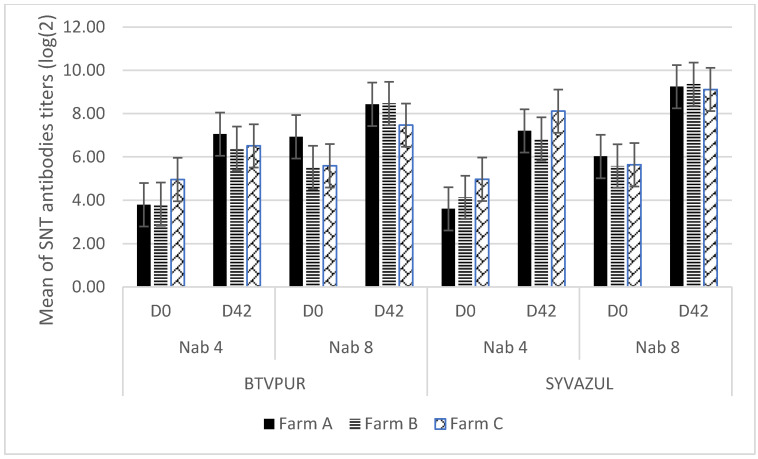
Evolution of neutralizing antibodies (Nab) in the three farms (A, B, C) according to the vaccine used as a booster (vertical bars represent ± 1 standard error). Nab 4: neutralizing antibodies against BTV-4; Nab 8: neutralizing antibodies against BTV-8; D0: day 0 prior to booster vaccination; D42: day 42 after booster vaccination.

**Table 1 viruses-14-02719-t001:** Study design (vaccinal interventions and sampling).

		Farm A (19 Cattle)	Farm B (18 Cattle)	Farm C (13 Cattle)
Vaccination	Primovaccination 1st injection (date-vaccine)	2-4-2021 BTVPUR^®^ 4 + 8	18-3-2021 BTVPUR^®^ 4 + 8	10-05-2021 BTVPUR^®^ 4 + 8
	Primovaccination 2nd injection (date-vaccine)	23-4-2021 BTVPUR^®^ 4 + 8	9-4-2021 BTVPUR^®^ 4 + 8	1-06-2021 BTVPUR^®^ 4+8
	Boost 2022 (nb cattle vaccinated with BTVPUR^®^ 4 + 8 or SYVAZUL BTV-4 + 8)	18-03-2022 (D0) (10 BTVPUR and 9 SYVAZUL)	18-3-2022 (D0) (9 BTVPUR and 9 SYVAZUL)	22-4-2022 (D0) (5 BTVPUR and 8 SYVAZUL)
Sampling	EDTA blood samples	D0, D10, D56	D0, D10, D56	D0, D10, D56
	Blood samples (serum)	D0, D42	D0, D42	D0, D42

D0: Day of vaccine booster; D10, 42, 56: days after the vaccine booster.

**Table 2 viruses-14-02719-t002:** Serological results (ELISA and SNT BTV4 and 8) from sera sampled at D0.

			Number of Positive Animals per Analysis		
	Number of Animals	VP7 cELISA	VP2 ELISA BTV-4	SNT4	SNT8
Farm A	19	12 (63.2)	14 (77.8 *)	14 (73.7)	19 (100)
Farm B	18	17 (94.4)	18 (100)	15 (83.3)	18 (100)
Farm C	13	10 (76.9)	7 (70 **)	12 (92.3)	12 (92.3)
Total	50	39 (78)	39 (84.8 ***)	41 (82)	49 (98)

In brackets: percentage of positive animals per farm and per analysis. For VP2 ELISA BTV-4 only 18 (*), 10 (**), and 46 (***) animals were tested.

**Table 3 viruses-14-02719-t003:** Average serum-neutralizing antibody (Nab) titer per farm and per serotype and difference between the average titers at D42 and D0 according to the vaccine used for boosting.

		Mean Titer at D0				Mean Titer at D42				Delta Mean Titer D42-D0			
		Farm A	Farm B	Farm C	General Mean	Farm A	Farm B	Farm C	General Mean	Farm A	Farm B	Farm C	All Farms
BTVPUR	Mean Nab 4	3.80 [2,32–5,32]	3.82 [2,32–5,91]	4.96 [3,32–7,91]	4.05	7.06 [5,32–8,91]	6.41 [5,32–7,32]	6.51 [5,32–7,91]	6.71	3.26	2.59	1.55	2.66
	Mean Nab 8	6.93 [5,32–8,32]	5.52 [3,32–6,91]	5.59 [3,91–6,91]	6.12	8.43 [7,91–9,91]	8.47 [7,91–9,32]	7.47 [6,32–8,91]	8.28	1.50	2.95	1.88	2.16
SYVAZUL	Mean Nab 4	3.61 [2,32–5,32]	4.14 [2,32–6,32]	4.97 [2,32–6,32]	4.21	7.20 [5,32–8,91]	6.83 [5,32–8,32]	8.12 [5,32–9,32]	7.29	3.59	2.69	3.15	3.08
	Mean Nab 8	6.03 [4,32–8,32]	5.58 [4,32–6,32]	5.64 [2,32–7,32]	5.75	9.25 [8,32–9,91]	9.36 [8,91–9,91]	9.12 [7,32–9,91]	9.26	3.22	3.78	3.48	3.51

[range of results].

## Data Availability

The data that support the findings of this study are available from the corresponding author upon reasonable request.

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
