# Peer review of "Serological Responses in Cattle following Booster Vaccination against Serotypes 4 and 8 Bluetongue Virus with Two Bivalent Commercial Inactivated Vaccines"

_viruses, 2022, doi:10.3390/v14122719_

Round 1

Reviewer 1 Report

The Authors describe a small-scale trial for exploring the feasibility and benefits of an heterologous prime/boost vaccination for BTV serotypes 4 and 8.

To be honest, I don't think there was any need for such work to evaluate this, as prime/boost vaccinations with different brands are used in the field, and no one has ever given importance to it.

Nonetheless, the study gives a scientific evidence for the feasibility and even the greater benefit of using different brands for prime and boost vaccination.  The consequences of such results for the "real world" will maybe be low, as the choice of vaccines mostly follows marketing rules. But here I am asked to gauge the paper froma  technical and scientific point of view, and the paper is technically and scientifically sound.

Minor comments and suggestions are present in the uploaded file.

Overall, it's my opinion that the manuscript has the the right length, without unnecessary redundancies. M&M is clear and easy to read. Results should be partially revised for clarity and correctness. Discussion covers all aspects of the Results. 

I didn't evaluate the statistics because it is beyond my expertise

Reviewer 2 Report

Dear Authors,

This manuscript presents a highly interesting and informative comparison of cattle serological responses to booster vaccination with two different bivalent inactivated BTV vaccines (BTVPUR and SYVAZUL) following prior vaccination with BTVPUR. The efficacy of booster responses was assessed to determine whether heterologous BTV vaccination is just as effective as homologous vaccination to potentially allow more flexibility during BTV vaccination campaigns. This study has wide impact to all countries which use such vaccines during BTV vaccination campaigns, and the study provides evidence to suggest that in cases where homologous vaccines are in short supply or unavailable, the use of a heterologous vaccines are not detrimental and in fact, in the case of neutralizing antibody responses, provide better boosting of the bovine immune response and may therefore be preferrable. This study is the first to compare serological responses in booster vaccination with bivalent vaccines (BTV-4 and BTV-8). I highly recommend publication in the journal, Viruses, following some minor modifications as detailed in the attached document.
